# Efficient conversion of chemical energy into mechanical work by Hsp70 chaperones

Salvatore Assenza[1,2†], Alberto Stefano Sassi[3,4†], Ruth Kellner[5], Benjamin Schuler[5,6], Paolo De Los Rios[3,7], Alessandro Barducci[8*]

[1]Laboratory of Food and Soft Materials, ETH Zürich, Zürich, Switzerland; [2]Departmento de Física Teórica de la Materia Condensada, Universidad Autónoma de Madrid, Madrid, Spain; [3]Institute of Physics, School of Basic Sciences, École Polytechnique Fédérale de Lausanne (EPFL), Lausanne, Switzerland; [4]IBM TJ Watson Research Center, Yorktown Heights, New York, United States; [5]Department of Biochemistry, University of Zurich, Zurich, Switzerland; [6]Department of Physics, University of Zurich, Zurich, Switzerland; [7]Institute of Bioengineering, School of Life Sciences, Ecole Polytechnique Fédérale de Lausanne (EPFL), Lausanne, Switzerland; [8]Centre de Biochimie Structurale (CBS), INSERM, CNRS, Université de Montpellier, Montpellier, France

**Abstract** Hsp70 molecular chaperones are abundant ATP-dependent nanomachines that actively reshape non-native, misfolded proteins and assist a wide variety of essential cellular processes. Here, we combine complementary theoretical approaches to elucidate the structural and thermodynamic details of the chaperone-induced expansion of a substrate protein, with a particular emphasis on the critical role played by ATP hydrolysis. We first determine the conformational free-energy cost of the substrate expansion due to the binding of multiple chaperones using coarse-grained molecular simulations. We then exploit this result to implement a non-equilibrium rate model which estimates the degree of expansion as a function of the free energy provided by ATP hydrolysis. Our results are in quantitative agreement with recent single-molecule FRET experiments and highlight the stark non-equilibrium nature of the process, showing that Hsp70s are optimized to effectively convert chemical energy into mechanical work close to physiological conditions.

**\*For correspondence:**
alessandro.barducci@cbs.cnrs.fr

[†]These authors contributed equally to this work

**Competing interests:** The authors declare that no competing interests exist.

## Introduction

Even though in vitro most proteins can reach their native structure spontaneously (*Anfinsen, 1973*), this is not always the case in cellular conditions and proteins can populate misfolded states which can form cytotoxic aggregates (*Dobson, 2003*). In order to counteract misfolding and aggregation, cells employ specialized proteins, called *molecular chaperones*, which act on non-native protein substrates by processes that stringently depend on ATP hydrolysis for most chaperone families (*Hartl, 1996*). Among them, the ubiquitous 70 kDa heat-shock proteins (Hsp70s) play a special role because they assist a plethora of fundamental cellular processes beyond prevention of aggregation (*Clerico et al., 2019*; *Rosenzweig et al., 2019*).

Decades of biochemical and structural studies have clarified the core elements of the Hsp70 functional cycle at the molecular level (*Mayer, 2013*). Hsp70s consist of two domains: the substrate binding domain (SBD) interacts with disparate substrate proteins, whereas the nucleotide binding domain (NBD) is responsible for the binding and hydrolysis of ATP. The two domains are allosterically coupled, and the nature of the nucleotide bound to the NBD affects the structure of the SBD

and as a consequence the affinity for the substrate and its association/dissociation rates. More precisely, when the chaperone is in the ATP-bound state, the SBD is open and easily accessible to the substrate, whereas the SBD is closed when ADP is bound. These structural differences result in substrate binding and unbinding rates when ATP is bound that are orders of magnitude faster than when ADP is bound (*Mayer et al., 2000*). Furthermore, the coupling is bidirectional: the substrate, together with a co-localized J-domain protein (JDP) that serves as cochaperone (*Kampinga and Craig, 2010*; *Kampinga et al., 2019*), greatly accelerates the hydrolysis of ATP. Substrate binding thus benefits from the fast association rate of the ATP-bound state and the slow dissociation rate of the ADP-bound state, resulting in a non-equilibrium affinity (*ultra-affinity*) that can be enhanced beyond the maximum limit allowed by thermodynamic equilibrium, namely the affinity of the ADP-bound state (*De Los Rios and Barducci, 2014*; *Barducci and De Los Rios, 2015*).

More recently, the consequences of Hsp70 binding on the conformational ensembles of its substrates have also been investigated. Several lines of evidence indicate that the binding of Hsp70s to a polypeptide induces its expansion. Biochemical assays revealed that binding of Hsp70 increases the sensitivity of misfolded Luciferase to proteolysis and decreases its propensity to bind Thioflavin-T, strongly suggesting a loss of compactness (*Sharma et al., 2010*). Nuclear Magnetic Resonance (NMR) measurements have shown that Hsp70s destabilize the tertiary structure of several different substrates (*Lee et al., 2015*; *Sekhar et al., 2015*). Moreover, a single-molecule study based on Förster resonance energy transfer (FRET) spectroscopy quantified the considerable expansion of unfolded rhodanese in native conditions upon binding of multiple Hsp70 chaperones. In particular, this study revealed that the expansion is stringently ATP-dependent, because upon ATP exhaustion the system relaxes to the expansion values observed in the absence of chaperones (*Kellner et al., 2014*).

Despite these advances in the characterization of Hsp70 functioning, the mechanistic understanding of how the energy of ATP hydrolysis is used to expand a substrate has lagged behind. Our goal here is precisely to fill this gap between the molecular and functional characterization of Hsp70. To this aim, we first explore the structural and energetic features of Hsp70-bound rhodanese using Molecular Dynamics (MD) simulations. We next integrate this molecular information into a rate model that explicitly includes the Hsp70-rhodanese interactions and the chaperone ATPase cycle, thus elucidating how Hsp70s convert the chemical energy of ATP into mechanical work necessary to expand their substrates.

## Results

### Structural and thermodynamic characterization of chaperone-substrate complexes

To characterize the main features of chaperone-induced expansion, we performed MD simulations of the Hsp70/rhodanese complexes. We relied on a one-bead-per-residue Coarse Grained (CG) force field (*Smith et al., 2014*), which has been tailored to match experimental FRET data of intrinsically disordered proteins and satisfactorily reproduces the compactness of unfolded rhodanese in native conditions without any further tuning (see Materials and methods). Hsp70 chaperones were modeled with a structure-based potential to account for their excluded volume and they were artificially restrained to binding sites on the substrate. We identified six binding sites on the rhodanese sequence using two distinct bioinformatic algorithms (*Rüdiger et al., 1997*; *Van Durme et al., 2009*). Considering that each binding site could be either free or bound to a Hsp70 protein, we thus took into account a total of $2^6$ = 64 distinct chaperone/substrate complexes, which were exhaustively simulated. In *Figure 1*, we report the distributions of the substrate potential energy and of the radius of gyration ($R_g$) for three representative complexes with one (left), three (center) and six (right) bound chaperones. Consistently with FRET results (*Kellner et al., 2014*), chaperone binding leads to larger radii of gyration and higher potential energies, implying that the excluded-volume interactions due to the large Hsp70s progressively expand the complex and disrupt the attractive intra-chain interactions in rhodanese.

We then calculated the conformational free energy of all the possible chaperone/rhodanese complexes to obtain a quantitative picture of the energy landscape governing the chaperone-induced expansion. To this aim, we performed extensive sets of non-equilibrium steering MD trajectories for

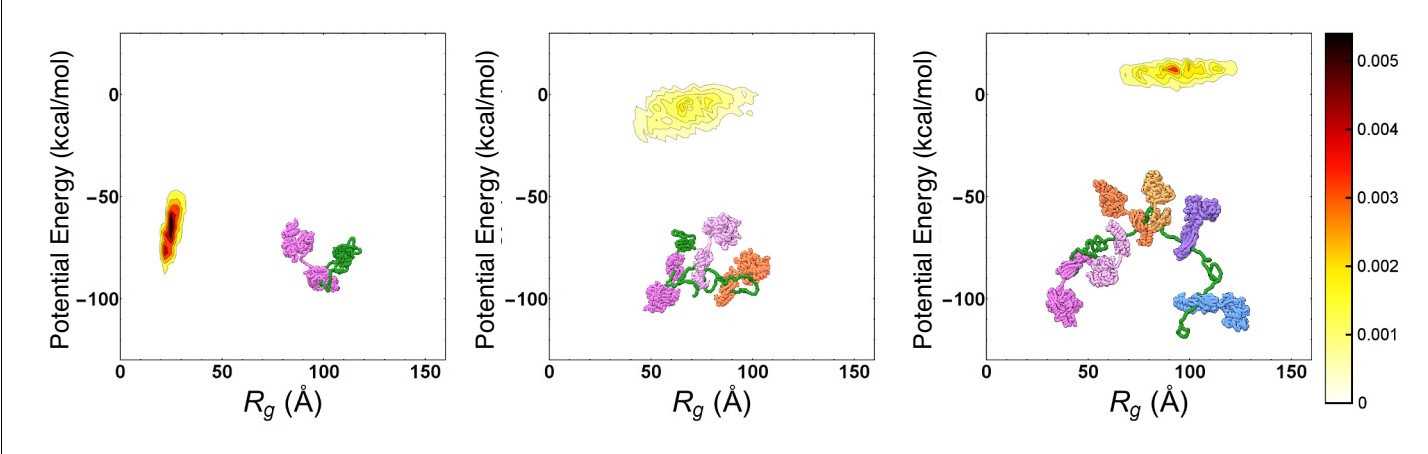

**Figure 1.** Progressive expansion of substrate by multiple Hsp70 binding. Probability density maps of substrate potential energy and radius of gyration for representative Hsp70/rhodanese complexes with one (left), three (center) and six (right) bound chaperones. The different Hsp70 chaperones have been represented with different colors to ease their discernibility.

The online version of this article includes the following source data for figure 1:

**Source data 1.** Text files containing the data used to generate the probability density maps in *Figure 1*.

each complex, and measured the work needed to steer it to a completely extended reference structure ($R_g$>260 Å), whose conformational free energy is not affected by chaperone binding. Equilibrium free-energy differences with respect to this reference state were then estimated from non-equilibrium work distributions via the Jarzynski equality (*Jarzynski, 1997*), thus allowing the determination of the conformational free energy $\Delta G$ of each distinct chaperone/substrate complex (see *Figure 2— figure supplement 1* and Materials and methods).

In *Figure 2* (main), we report $\Delta G$ for each complex as a function of its mean radius of gyration using different colors for different stoichiometries. The conformational free energy increased with the swelling of the substrate due to the progressive binding of the chaperones. The increase in substrate potential energy due to the loss of intra-chain interactions upon Hsp70 binding is therefore only marginally compensated by the gain in conformational entropy. Notably, the conformational free energy is not uniquely determined by the stoichiometry, and is significantly affected by the specific binding pattern. The conformational free-energy cost $\Delta\Delta G$ of adding a single chaperone (inset in *Figure 2*) is positive for all complexes, but it varies from 2 kcal/mol up to 7 kcal/mol depending on the stoichiometry of the complex and on the particular choice of the binding sites. The increase of $\Delta G$ as a function of $R_g$ is quantitatively captured by *Sanchez (1979)* theory for the coil-to-globule collapse transition in polymers (see *Figure 2* and Materials and methods). Remarkably, the excellent agreement is not the outcome of a fitting procedure since all the parameters were extracted from experiments (see Appendix 2). This result further reinforces the reliability of our simulations as well as the general applicability of the present setup beyond the particular system considered in this work.

## ATP hydrolysis promotes multiple chaperone binding

The structural and thermodynamic characterization obtained by molecular simulations can be profitably complemented by a kinetic model encompassing relevant biochemical processes in order to determine the probability of each chaperone/substrate complex as a function of the chemical conditions. Notably, a model of the Hsp70 biochemical cycle based on experimental rates was previously used to illustrate how ATP-hydrolysis may result into non-equilibrium ultra-affinity for peptide substrates (*De Los Rios and Barducci, 2014*). Here, we extend this result to the more complex case of Hsp70-induced expansion by taking into account multiple chaperone binding events and their consequences on the conformational free energy of the substrate.

In our model, each state corresponds to a single configuration of the chaperone/substrate complex, which is defined by the occupation state of the six Hsp70 binding sites on rhodanese. Each site

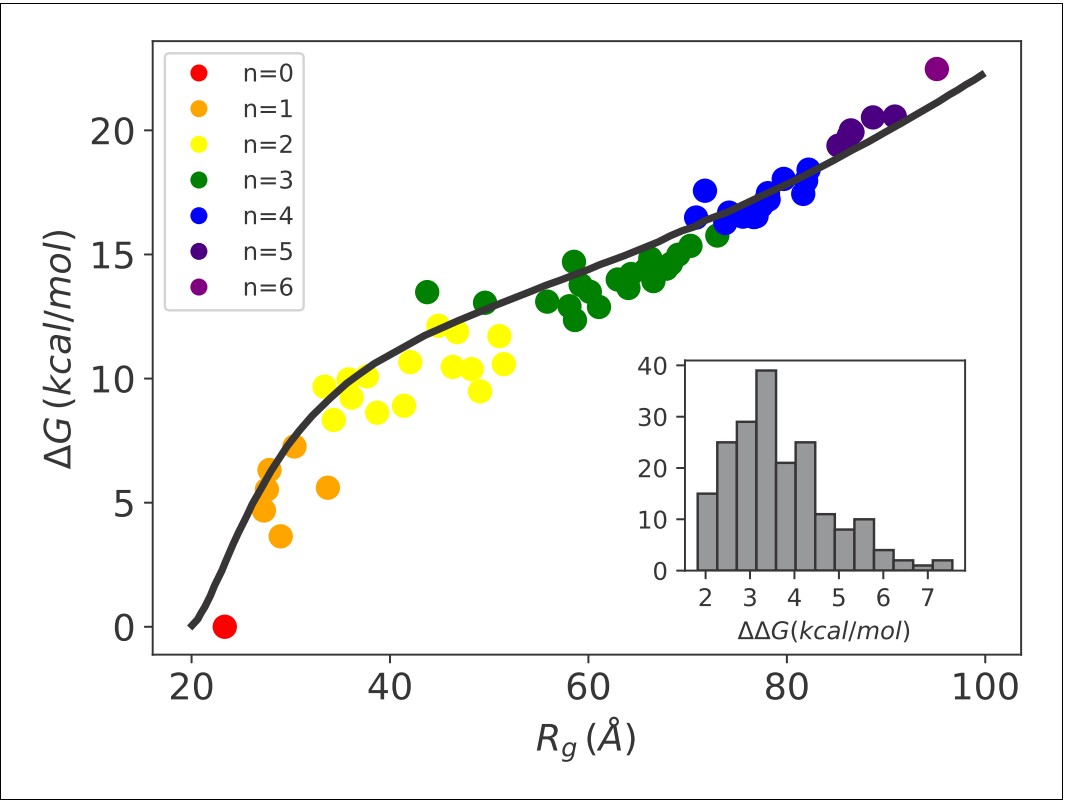

**Figure 2.** Hsp70-induced swelling increases the conformational free energy of the system. Conformational free-energy differences $\Delta G$ of the Hsp70/rhodanese complexes with respect to the unbound substrate (n = 0) plotted as a function of the corresponding radius of gyration $R_g$. Each point represents one of the 64 possible binding configurations with color code indicating the number of bound chaperones. The black curve was obtained using the model in *Sanchez (1979)* (see Appendix 2). (inset) Distribution of $\Delta\Delta G$ corresponding to the free-energy cost for binding an additional Hsp70 to a chaperone/substrate complex.

The online version of this article includes the following source data and figure supplement(s) for figure 2:

**Source data 1.** Data from simulation results and Sanchez theory used to generate the plot in *Figure 2* and the histogram in the inset of *Figure 2*.

**Figure supplement 1.** Free energy computation of different combinations of bound chaperones from steered MD simulations.

**Figure supplement 1—source data 1.** Simulation data used to generate the free energy plots; the work performed by the pulling force for each trajectory is also included in the subfolders.

can be either free or occupied by an ADP- or ATP-bound chaperone for a total of $3^6$ = 729 different states. All the relevant molecular processes corresponding to transitions between these states are explicitly modeled, including chaperone binding/unbinding, nucleotide exchange and ATP hydrolysis (see *Figure 3*). We took advantage of available biochemical data for determining the rate constants associated to all the relevant reactions (see Materials and methods). Importantly, kinetic rates for Hsp70 binding were modulated by the conformational free energies determined by CG MD simulations. Indeed, the unbinding rates of Hsp70 from large-sized protein substrates were observed to be similar to the ones from small peptides, whereas the binding rates can be up to two orders of magnitude smaller (*Schmid et al., 1994*; *Mayer et al., 2000*; *Kellner et al., 2014*). This evidence was further corroborated by a recent NMR study (*Sekhar et al., 2018*) suggesting a conformational selection scenario where the energetic cost due to substrate expansion mostly affects the Hsp70/rhodanese binding rate. Accordingly, we thus considered a substrate-independent unbinding rate constant $k_{off}$, while we expressed the binding rate constant as

$$k_{on,ij} = k_{on}^0 \exp[-\beta\Delta\Delta G_{ij}],\qquad(1)$$

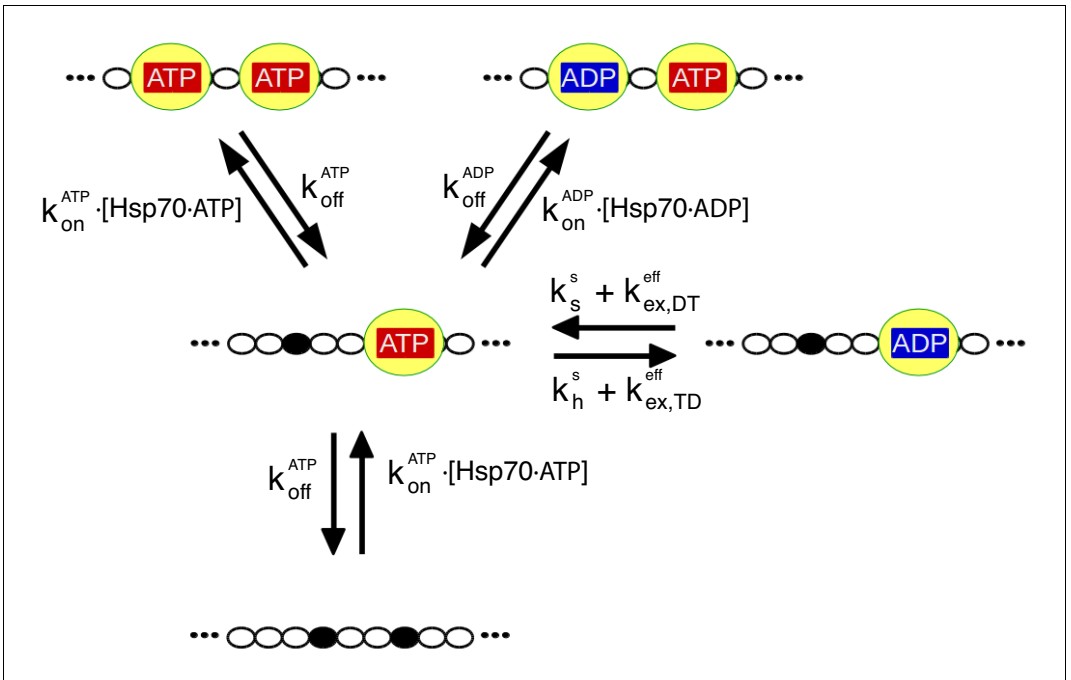

**Figure 3.** The relevant biochemical reactions of the Hsp70/rhodanese system can be described by a rate model. Each chaperone binding site on rhodanese (black dots) can be either free or occupied by an Hsp70 (yellow), which in turn can be either ADP- or ATP-bound. We depict here for the sake of clarity only a representative portion of the full model, which takes into account six binding sites. The reaction cycle is governed by the rates for chaperone binding/unbinding to the substrate ($k_{on}^{ATP}, k_{on}^{ADP}, k_{off}^{ATP}, k_{off}^{ADP}$) and for hydrolysis ($k_h^s$), synthesis ($k_s^s$) and exchange ($k_{ex,DT}^{eff}, k_{ex,TD}^{eff}$) of nucleotides bound to the chaperones (see Materials and methods for further details). Importantly, the binding rate constants, $k_{on}^{ATP}$ and $k_{on}^{ADP}$, take into account the conformational free energies, according to *Equation (1)*.

where $\beta = 1/k_B T$, $k_B$ is the Boltzmann constant, $T$ is the absolute temperature, $k_{on}^0$ is the binding rate measured for a peptide substrate, and $\Delta\Delta G_{ij}$ is the conformational free-energy cost of Hsp70 binding, which depends on the specific initial and final binding patterns $i$ and $j$ in the rhodanese/chaperone complex (see *Figure 2*, inset). The interactions with JDP cochaperones were not explicitly modeled but the cochaperones were assumed to be colocalized with the substrate, so that their effect was implicitly taken into account in the choice of the rate constants for ATP hydrolysis (*Kampinga and Craig, 2010*; *Hu et al., 2006*).

The analytical solution of the model provides the steady-state probability of each binding configuration and allows the exploration of their dependence on external conditions. It is particularly instructive to investigate the system behavior as a function of the ratio between the concentrations of ATP and ADP, which is intimately connected to the energy released by ATP hydrolysis. At thermodynamic equilibrium, the $[ATP]/[ADP]$ ratio is greatly tilted in favor of ADP ($[ATP]_{eq}/[ADP]_{eq} \simeq 10^{-9} - 10^{-8}$; *Alberty, 2005*), whereas in the cell ATP is maintained in excess over ADP by energy-consuming chemostats ($[ATP]/[ADP] > 1$; *Milo and Phillips, 2015*). The $[ATP]/[ADP]$ ratio hence determines how far the system is from equilibrium, thus representing a natural control parameter for the non-equilibrium biochemical cycle. We thus report in *Figure 4* (top panel) the compound probabilities for complexes with the same stoichiometry $n$ as a function of this nucleotide ratio. In conditions close to equilibrium (very low values of $[ATP]/[ADP]$), the vast majority of the substrate proteins are free and only about 10% of them are bound to a single chaperone. The population of equimolar complexes increases for $[ATP]/[ADP]$ between $10^{-2}$ and $10^{-1}$ and gives way to larger complexes with multiple chaperones for higher values of the nucleotide ratio. For $[ATP]/[ADP] > 1$, most substrates are bound to at least 4 chaperones, with an average stoichiometry $<n> \sim 4.9$ (solid line in *Figure 4*), bottom panel). Further increase of the nucleotide ratio does not

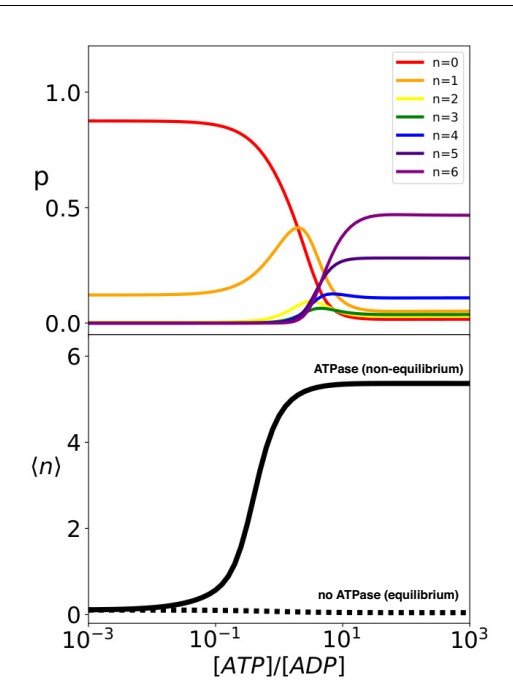

**Figure 4.** Hsp70 binding is a non-equilibrium process that depends on ATP hydrolysis. (Top) compound probabilities for Hsp70/substrate complexes with given number of bound chaperones $n$ as a function of [ATP]/[ADP]. (Bottom) Mean value $<n>$ as a function of [ATP]/[ADP] with (solid line) and without (dashed line) ATP hydrolysis.

The online version of this article includes the following source data for figure 4:

**Source data 1.** Data from rate model employed to generate the plots in *Figure 4*.

significantly change this scenario indicating an almost constant behaviour in large excess of ATP ($[ATP]/[ADP]>10$). It is important here to underscore that the binding of the chaperones in these conditions is a non-equilibrium effect, driven by the Hsp70-induced hydrolysis of ATP, and it is not a mere consequence of the excess of ATP over ADP or of the large difference between the substrate association rates to the ATP- and ADP-bound chaperones. Indeed, if we neglect Hsp70 ATPase activity ($k_h^h, k_s^h = 0$) without changing any of the other model parameters, efficient chaperone binding is abolished ($<n> \ll 1$, as shown in the bottom panel of *Figure 4*, dashed line). As a matter of fact, in such equilibrium scenario an excess of ATP over ADP actually slightly disfavors chaperone binding, because the Hsp70 affinity for the substrate is slightly lower in the ATP-bound state than in the ADP-bound state ($k_{off}^{ATP}/k_{on}^{ATP}>k_{off}^{ADP}/k_{on}^{ADP}$, see *De Los Rios and Barducci, 2014* for further discussion.)

Combining the steady-state probabilities derived from the rate model with the results of the MD simulations, we can now exhaustively characterize the structural properties of the system. This provides the opportunity to directly compare our model with the results from FRET experiments both in equilibrium and non-equilibrium conditions. To this aim, we first focused on the average radius of gyration of the system at thermodynamic equilibrium ($[ATP] \ll [ADP]$) or in non-equilibrium conditions with ATP in large excess over ADP ($[ATP]/[ADP]>10$). In order to probe the robustness of our results with respect to inaccuracies in the molecular model, we also took into account normally distributed errors on the conformational free energies $\Delta G_i$.

The results are reported as histograms in *Figure 5* and they suggest that at equilibrium the average radius of gyration is extremely close to what would be measured in the case of free substrate (dashed line). This is in agreement with the experimental observation that the formation of rhodanese–DnaK complexes is strictly dependent on the hydrolysis of ATP and that ADP cannot trigger the expansion of the substrate (*Kellner et al., 2014*). Conversely, in large excess of ATP we observe a substantial swelling of the substrate ($75< R_g <95$ Å) due to the ultra-affine binding of Hsp70s. This finding is fully compatible with the size of DnaK/DnaJ/rhodanese complexes determined by single-molecule FRET experiments in excess of ATP (*Kellner et al., 2014*). In this regime, the limited effects of cochaperone binding on substrate conformations, which are not explicitly included in the model, play a minor role in determining the global expansion of the complex.

A more quantitative comparison between the model and the FRET results can be achieved by back-calculating the transfer efficiencies that were experimentally measured for five distinct pairs of fluorescent dyes (*Kellner et al., 2014*). In equilibrium conditions, namely when $[\mathrm{ATP}]/[\mathrm{ADP}] \ll 1$, the calculated FRET efficiency is ~0.8 for all considered pairs of fluorescent dyes (inset of *Figure 5*, blue circles) and it matches the experimental results for the compact unbound rhodanese (~0.8). A dramatic difference is instead observed in excess of ATP (red circles), where the expansion of the substrate leads to a significant decrease of the calculated efficiency , in excellent agreement with the experimental values measured in similar conditions (black circles). Remarkably, the results

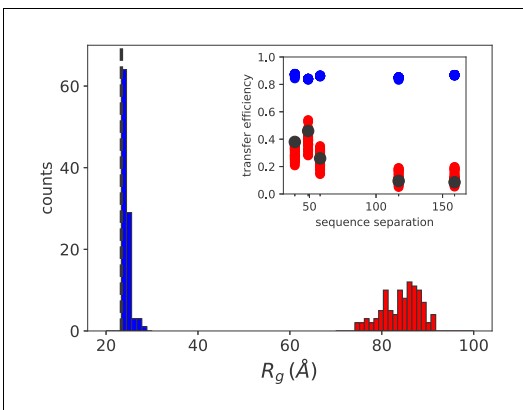

**Figure 5.** Simulation results match sm FRET experimental data. Histograms of the radius of gyration for equilibrium (blue) and non equilibrium (red) values of [ATP]. The black dashed line indicates the average radius of unbound rhodanese. (inset) FRET transfer efficiencies as a function of the sequence separation between the fluorescent dyes. The black circles correspond to the experimental values (*Kellner et al., 2014*). Calculated efficiencies taking into account uncertainties are reported as blue (equilibrium conditions) and red circles (ATP excess).

The online version of this article includes the following source data for figure 5:

**Source data 1.** Molecular simulation data used to generate the plots in *Figure 5*.

correctly captured the non-monotonic behaviour of FRET efficiency as a function of the sequence separation between the dyes, which was not reproduced in previous calculations (*Kellner et al., 2014*). This agreement corroborates the prediction of the DnaK binding sites on the rhodanese sequence and the overall reliability of our model.

## Energy balance and thermodynamic efficiency

Molecular chaperones consume energy via ATP hydrolysis in order to expand rhodanese. It is hence important to determine how effective they are as molecular machines, as well as to assess how favorable the physiological conditions are to perform their biological task.

To this aim, we calculated the global increase in the overall conformational free energy of the substrate with respect to equilibrium conditions, $\Delta G_{Swell}$ (*Figure 6*, top). This quantity measures the excess probability of each complex with respect to equilibrium conditions weighted by its corresponding conformational free energy $\Delta G_i$.

$$\Delta G_{Swell} = \sum_{i} \left[ p_i \left( \frac{[ATP]}{[ADP]} \right) - p_i^{eq} \right] \Delta G_i, \qquad (2)$$

where $p_i \left( \frac{[ATP]}{[ADP]} \right)$ is the probability of complex $i$ for a given value of $[ATP]/[ADP]$ and $p_i^{eq}$ is the same quantity computed at equilibrium conditions. In order to investigate the conversion of chemical energy into mechanical work, it is instructive to focus on the ratio between $\Delta G_{Swell}$ and the free energy of hydrolysis of ATP $\Delta G_h$,

$$\Delta G_h = k_B T \left[ \ln \left( \frac{[ATP]}{[ADP]} \right) - \ln \left( \frac{[ATP]_{eq}}{[ADP]_{eq}} \right) \right]. \qquad (3)$$

The ratio $\Delta G_{Swell}/\Delta G_h$ reports on the effectiveness of the transduction process. We plot in *Figure 6* (top) this quantity as a function of the $[ATP]/[ADP]$ ratio considering the estimated inaccuracies of the model as previously done for the gyration radius. Not surprisingly, all these curves exhibit a maximum because the probabilities of the different states, and thus also $\Delta G_{Swell}$, attain plateaus for $[ATP] \gg [ADP]$ (see 4, top panel), whereas $\Delta G_h$ increases monotonically with the nucleotide ratio. The regime where transduction is maximally efficient intriguingly corresponds to values of $[ATP]/[ADP]$ that are typical of cellular conditions (grey area).

We highlight that in our model Hsp70 functioning encompasses two distinct yet intertwined processes: the ATP-dependent binding of the chaperones to the substrate, and its consequent expansion. In this two-step mechanism, the amount of energy available for the mechanical expansion is limited by that provided by non-equilibrium Hsp70 binding, which does not explicitly depend on the overall conformational properties of the substrate. To further dissect the energetic determinants of Hsp70 functioning and obtain more general conclusions, we thus analyzed the energy balance of chaperone binding to a model substrate, such as a peptide with a single binding site. To this aim, we focused on a simplified reaction cycle, which essentially corresponds to a single triangle within the overall scheme in *Figure 3* and does not imply any conformational free-energy variation upon Hsp70 binding. We report in *Figure 6* (bottom panel, black solid line) the non-equilibrium dissociation constant, $K_d^{neq}$, normalized with respect to its equilibrium value $K_d^{eq}$, as a function of [ATP]/[ADP].

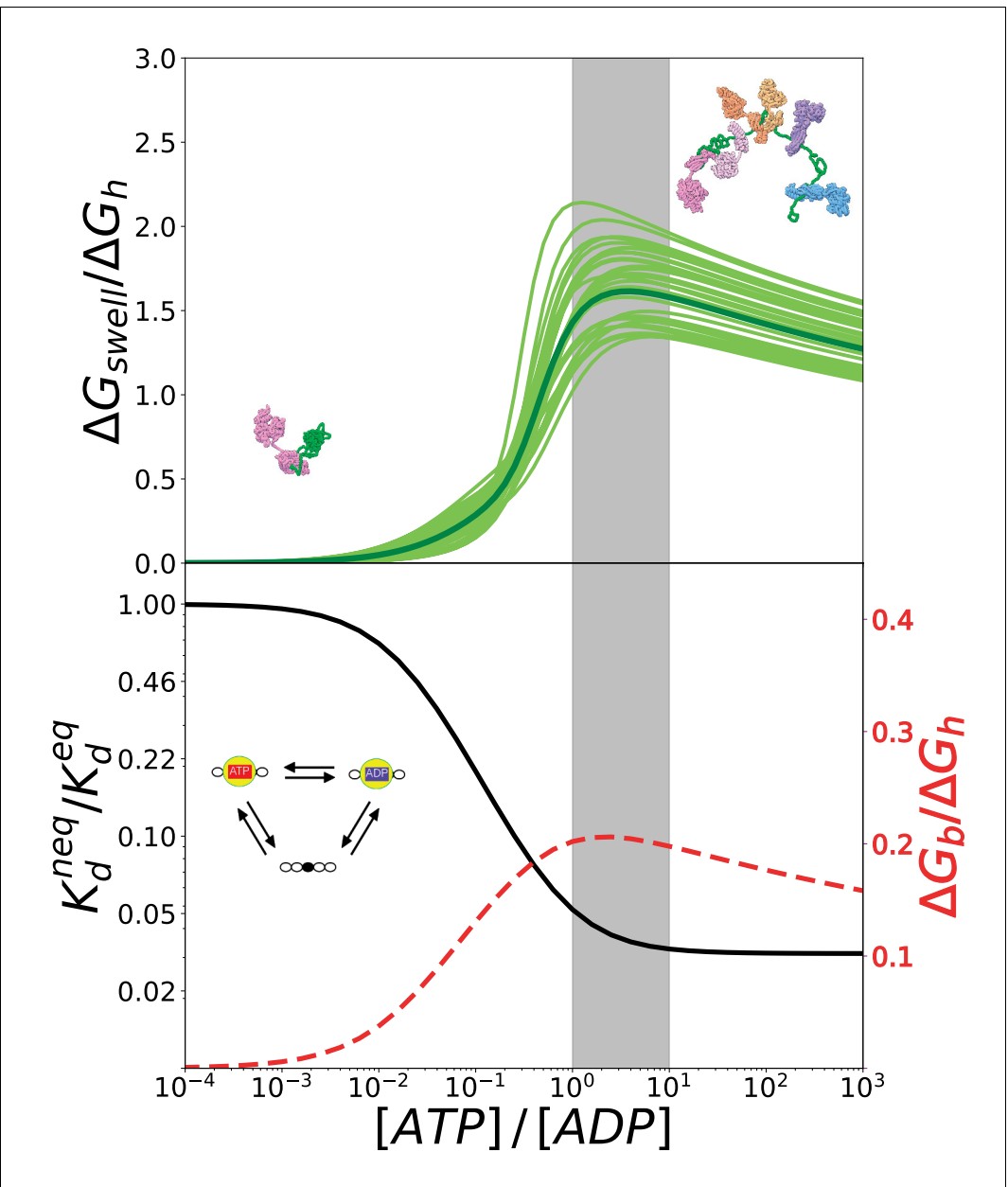

**Figure 6.** The conversion of chemical energy into mechanical work by Hsp70s is optimal in cellular conditions thanks to the chaperone intrinsic rates. (Top) Ratio between the conformational free energy and the free energy of ATP hydrolysis, as a function of [ATP]/[ADP]. Dark green curve results from data from molecular simulations and light green curves takes into account normally-distributed uncertainties on calculated $\Delta G_i$. (Bottom) Effective dissociation constant in the case of a single binding site normalized with respect to the corresponding value in equilibrium, as a function of [ATP]/[ADP] (solid black line). Ratio between the non-equilibrium excess of binding free energy $\Delta G_b$ and the free energy of ATP hydrolysis $\Delta G_h$, as a function of [ATP]/[ADP] (dashed red line). The gray region indicates the interval corresponding to physiological conditions.

The online version of this article includes the following source data for figure 6:

**Source data 1.** Free-energy curves used to generate the plots in *Figure 6*.

When the ratio between the concentrations of ATP and ADP approaches the physiological regime, the dissociation constant drops significantly until it settles at a value that is two orders of magnitude lower than its equilibrium counterpart, as already discussed in *De Los Rios and Barducci (2014)*.

Here, we convert the dissociation constant into a binding free-energy excess with respect to equilibrium

$$\Delta G_b = -k_B T \ln\left[\frac{K_d^{neq}}{K_d^{eq}}\right] \tag{4}$$

that we can compare to the free energy of ATP hydrolysis, $\Delta G_h$, as previously done in the case of $\Delta G_{Swell}$. Interestingly, also in this case the energy ratio is maximal in cellular conditions (red dashed line in *Figure 6*, bottom panel), suggesting that the optimality of the overall expansion process does not depend on specific features of the substrate but it is a direct consequence of the intrinsic kinetic parameters of Hsp70 chaperones.

## Discussion

Integrating molecular simulations, polymer theory, single-molecule experimental data and non-equilibrium rate models, we have developed a comprehensive framework that provides a quantitative picture of Hsp70-induced expansion of substrate proteins and offers a broad insight into the cellular functioning of this versatile chaperone machine.

We relied on molecular simulations for characterizing the structural and thermodynamic features of the complexes formed by the bacterial chaperone DnaK and its unfolded substrate rhodanese. Notably, we investigated a large variety of possible chaperone-substrate complexes for determining their conformational free energy as a function of stoichiometry and chaperone binding patterns. This computational strategy based on an enhanced-sampling protocol confirmed that excluded-volume interactions upon chaperone binding can greatly perturb the conformational ensemble of the unfolded substrate leading to its expansion. Remarkably, simulation results were found to be in excellent agreement with the predictions of Sanchez theory for globule to coil transition, thus providing another example of how polymer theory can be successfully used to decipher the behaviour of disordered proteins (*Sherman and Haran, 2006*; *Hofmann et al., 2012*; *Schuler et al., 2016*). We then combined conformational free energies with available biochemical data to develop an analytical rate model of the chaperone/substrate reaction cycle, which included both chaperone binding/unbinding and nucleotide hydrolysis/exchange processes.

This model fully takes into account non-equilibrium effects due to ATP hydrolysis and represents a natural extension of the ultra-affinity framework originally developed for peptide substrates with a single Hsp70 binding site (*De Los Rios and Barducci, 2014*). We could thus investigate the population of each complex and the average structural properties of the system as a function of the ATP/ADP nucleotide ratio, which measures how far the system is from thermodynamic equilibrium. The reliability of the model was corroborated by a quantitative comparison with recent single-molecule FRET data, indicating that our non-equilibrium framework accurately captures the salient features of the ATP-dependent expansion. We then used this unprecedented access to the thermodynamics details of this complex molecular process to compare the free-energy cost associated with substrate swelling with the chemical energy released by ATP-hydrolysis. Remarkably, this analysis revealed that energy transduction is maximally efficient for ATP/ADP values in cellular conditions. This result hints at the possibility that Hsp70 chaperones have been tuned by evolution to optimize the conversion of chemical energy into mechanical work for substrate expansion. Further analysis indicated that this optimality is likely inherited from the intrinsic properties of Hsp70 chaperones, which can convert up to 20% of the ATP chemical energy into non-equilibrium, excess binding energy at physiological conditions (*Figure 6*, bottom panel).

From a broader perspective, the ATP-driven action of Hsp70s induces a *non-equilibrium* redistribution of their protein-substrates over their structural ensemble. In particular, thanks to the fine-tuning of the process by co-chaperones (J-domain proteins and Nucleotide Exchange Factors), the expansion process highlighted here, followed by substrate release, may result in the enhancement of the native state population beyond the predictions of thermodynamic equilibrium, as recently observed even under otherwise denaturing conditions (*Goloubinoff et al., 2018*). Consistently, Zhao and coworkers have recently observed that Hsp70 chaperones crucially contribute in vivo to the solubility and functionality of a sizeable fraction of the *E. coli* proteome that, in their absence, would instead spontaneously misfold and aggregate (*Zhao et al., 2019*). Remarkably, a similar effect has

been observed in vitro for the GroEL chaperonin (*Chakrabarti et al., 2017*; *Goloubinoff et al., 2018*), hinting at the possibility that multiple chaperone families might reshape the equilibrium conformational distribution of proteins through energy-consuming processes. These considerations might have important consequences for our ability to translate results from in vitro experiments to the active cellular context (*Bershtein et al., 2013*). Likewise, they raise fundamental questions about the evolution of protein sequences: indeed, since chaperones are ubiquitous and very much conserved across the different kingdoms of life, their ability to favor native states might have partially relieved the selection pressure for strong *equilibrium* thermodynamic stability, thus allowing evolution to proceed faster and to be more tolerant for slightly destabilizing mutations, as suggested in *Rutherford and Lindquist (1998)*; *Tokuriki and Tawfik (2009)*.

Besides the unfolding of non-native substrates discussed in this work, Hsp70s are highly versatile machines that play a fundamental role in a variety of diverse cellular functions such as protein translocation, protein translation, and disassembly of protein complexes. All these processes share basic analogies from the mechanistic point of view: in all these cases, Hsp70 binding to flexible substrates in constrained environments requires the energy of ATP hydrolysis (ultra-affinity) and results in the generation of effective forces due to excluded volume effects (entropic pulling), which ultimately drive protein translocation into mitochondria (*De Los Rios et al., 2006*; *Assenza et al., 2015*), clathrin cage disassembly (*Sousa et al., 2016*) and/or prevention of ribosome stalling (*Liu et al., 2013*). Here, by detailing how energy flows from ATP hydrolysis to mechanical work due to entropic pulling, we have elucidated a general force-generating mechanism of Hsp70 chaperones. This mechanism does not rely on any power-stroke conformational change but it rather depends on the efficient conversion of ATP chemical energy into ultra-affinity.

## Materials and methods

### Molecular model

In all the simulations, rhodanese and Hsp70 were coarse grained at the single-residue level as collections of beads centered on the $C_\alpha$ atom of each amino acid. The unfolded state of bovine rhodanese (PDB:2RHS) was modeled according to the force field for disordered proteins from *Smith et al. (2014)*. Two- and three-body bonded interactions along the substrate backbone were included via harmonic potentials, namely $V_\text{bond} = k_l \sum_b (r_b - l)^2 / 2$ and $V_\text{bend} = \frac{1}{2} k_\theta \sum_\alpha (\theta_\alpha - \theta_0)^2$, respectively. In the previous formulas, $r_b$ denotes bond lengths; $\theta_\alpha$ the bend angles; $l = 3.9$ Å; $(k_l/k_B T)^{-1/2} = 0.046$ Å; $\theta_0 = 2.12$ rad; $(k_\theta/k_B T)^{-\frac{1}{2}} = 0.26$; and $k_B T$ is the thermal energy. Four-body bonded interactions were implemented as Fourier terms, $V_\text{dihed} = k_B T \sum_d \sum_{s=1}^4 [A_s \cos(s\phi_d) + B_s \sin(s\phi_d)]$, where $\phi_d$ is the torsion angle and $A_1 = 0.705$, $A_2 = -0.313$, $A_3 = -0.079$, $A_4 = 0.041$, $B_1 = -0.175$, $B_2 = -0.093$, $B_3 = 0.030$, $B_4 = 0.030$. The steric repulsion was implemented through a Weeks-Chandler-Andersen potential, $V_{WCA} = \sum_{ij} V_r$, where

$$V_r = \begin{cases} 4k_B T \left[ \left(\frac{\sigma}{r_{ij}}\right)^{12} - \left(\frac{\sigma}{r_{ij}}\right)^6 \right] + k_B T & \text{if } r_{ij} \leq 2^{\frac{1}{6}}\sigma \\ 0 & \text{otherwise} \end{cases}. \tag{5}$$

In the previous formula, $r_{ij}$ is the distance between beads $i$ and $j$, while $\sigma = 4.8$ Å. The hydrophobic part of the potential is specific to the interacting residues and is modeled as the attractive part of the Lennard-Jones potential, $V_\text{hydro} = \epsilon_h \sum_{ij} V_h$, where

$$V_h = \begin{cases} 4\epsilon_{ij} \left[ \left(\frac{\sigma}{r_{ij}}\right)^{12} - \left(\frac{\sigma}{r_{ij}}\right)^6 \right] & \text{if } r_{ij} \geq 2^{\frac{1}{6}}\sigma \\ -\epsilon_{ij} & \text{otherwise} \end{cases}. \tag{6}$$

In the previous formula, $\epsilon_h = 0.7722 \, k_B T$ sets the overall strength of the hydrophobic interactions, while $\epsilon_{ij}$ depends on the residues $i$ and $j$ involved in the interaction, and is defined as the geometric mean of their hydrophobicities, $\epsilon_{ij} \equiv \sqrt{\epsilon_i \epsilon_j}$. The values of the hydrophobicities considered are based on a shifted and normalized Monera hydrophobicity scale (*Smith et al., 2014*). Electrostatic interactions were neglected based on control FRET experiments (see section 5.3). Without further tuning,

this force field gives a radius of gyration of unbound rhodanese equal to $R_g = (23.3 \pm 0.1)$ Å, which is in good agreement with the experimental value $R_g = (20.1 \pm 0.8)$ Å (*Kellner et al., 2014*; *Hofmann et al., 2014*).

We modeled Hsp70 by means of a simple structure-based potential (*Assenza et al., 2015*) built on the conformation of ADP-bound Hsp70 (*Bertelsen et al., 2009*, PDB:2KHO). We described both the NBD (residues 4–680) and the SBD (residues 690–603) as rigid bodies whereas the interdomain linker (residues 681–689) was modeled according to the potential for flexible proteins described above. Importantly, non-bonded interactions of Hsp70 residues were limited to excluded volume effects and described by WCA potential (see *Equation (5)*). Electrostatic interactions were not explicitly included due to their marginal role in Hsp70/rhodanese complexes evidenced by FRET experiments (Appendix 1).

The binding sites for DnaK on the substrate were identified by applying the algorithms by *Rüdiger et al. (1997)* and *Van Durme et al., 2009* on rhodanese and selecting only fragments for which at least partial consensus between the two predictions was obtained. Following this proce-dure, we identified six binding sites roughly centered on residues 10, 118, 131, 162, 188, 260 of the rhodanese sequence. The residues of the binding site were aligned to a SBD-bound peptide reported in the literature (PDB:1DKX *Zhu et al., 1996*) and constrained to move rigidly with the cor-responding SBD, thus ensuring that each chaperone was irreversibly bound to the substrate. Follow-ing this procedure, $2^6 = 64$ different chaperone/substrate complexes were built depending on the occupancy of each binding site.

## Simulation protocols

All the simulations were performed with a version of LAMMPS (*Plimpton, 1995*) patched with the open-source, community-developed PLUMED library (*Bonomi et al., 2019*), version 2.1 (*Tribello et al., 2014*). The temperature $T = 293$ K was controlled through a Langevin thermostat with damping parameter 16 ns$^{-1}$. The time step was set equal to 1 fs, and each residue had a mass equal to 1 Da.

In order to obtain conformational properties, for each of the 64 chaperone/rhodanese complexes we performed at least 10 independent simulations of $2 \cdot 10^7$ timesteps. To ensure that full equilibra-tion was achieved, only the last $10^7$ timesteps of the obtained trajectories were considered for analy-sis. Statistical errors on the computed quantities were estimated as standard errors of the mean computed across independent realizations and are smaller than the size of symbols reported in the figures. The FRET efficiency $E$ for a given couple of dyes was computed starting from the distance $r$ separating the corresponding amino acids as

$$E = \frac{1}{1 + \left(\frac{r}{r_0}\right)^6}, \tag{7}$$

where $r_0 = 54$ Å, as in *Kellner et al. (2014)*. For each realization, the time average of $E$ was com-puted. The final values employed to compute the results reported in the inset of *Figure 5* in the main text were obtained as the average between independent realizations.

The conformational free energies were computed by means of steered simulations, where for each complex rhodanese was pulled from equilibrium until an elongated conformation was obtained (*Figure 2—figure supplement 1*, top panel). Due to the large intermolecular distances, the effect of chaperones on the conformational properties of fully-stretched rhodanese is negligible, so that this state can be used as a reference to compute the free-energy differences between different chaper-one/rhodanese complexes. The pulling was implemented by adding a harmonic potential acting on the radius of gyration $R_g$ of rhodanese. The equilibrium position of the harmonic trap was increased at a constant pulling speed v = 10$^{-5}$Å/fs from the equilibrium value up to $R_g = R_g^{\text{fin}}$. For each chaper-one/substrate complex, 100 independent pulling simulations were performed, starting from uncorre-lated initial snapshots extracted from the equilibrium distribution. For each realization, the work $W$ performed by the bias potential during the steering process was measured. The free-energy differ-ence $\delta G$ between the equilibrium starting point and the reference state (corresponding to $R_g^{\text{fin}}$) was then computed via the *Jarzynski (1997)* equality:

$$e^{-\frac{\delta G}{k_B T}} = <e^{-\frac{W}{k_B T}}>, \tag{8}$$

where $<\ldots>$ denotes statistical average. The error on $\delta G$ was estimated according to the bootstrap method. The quantity $\Delta G$ considered in the main text was finally computed as $\Delta G = \delta G_0 - \delta G$, where $\delta G_0$ corresponds to the case of rhodanese alone (*Figure 2—figure supplement 1*, bottom panel). The uncertainty on $\Delta G$ was estimated by propagating the error bars on $\delta G$ and is always smaller than the size of symbols. In order to enhance the robustness of the results, the final values reported in the main text were obtained as a further average over the values of $R_g^{\mathrm{fin}}$ within the range 260 Å $\leq$ $R_g^{\mathrm{fin}} \leq$ 290 Å.

## Rate model

For the kinetic model we consider a system in which each of the six binding sites can either be occupied by a chaperone in the ATP or ADP state, or it can be free, so that in total there are $3^6 = 729$ possible configurations. The concentration $c_i$ of each state evolves in time according to a system of rate equations

$$\frac{dc_i}{dt} = \sum_j k_{ji} c_j - \sum_j k_{ij} c_i \tag{9}$$

where $k_{ij}$ is the transition rate from state $i$ to state $j$. The first term in the right hand side (*r.h.s.*) of *Equation (9)* represents the total flux of molecules from the other states toward state $i$, while the second term in the r.h.s. of *Equation (9)* accounts for the flux of molecules from state $i$ to any other state. We focused on the steady-state, when the concentrations of the various states do not change over time, which is defined by

$$\frac{dc_i}{dt} = 0 \tag{10}$$

Here, we provide a list of the relevant reactions that must be taken into account and of their corresponding rates. Each configuration is labelled by means of six symbols: 0 for empty sites, $T$ for sites occupied by an ATP-bound chaperone and $D$ for sites occupied by an ADP-bound chaperone (*e.g.* $(0,T,0,D,0,0)$, where the first, third, fifth and sixth Hsp70 binding sites are unoccupied, the second binding site is associated to a chaperone in the ATP-bound state while the fourth binding site is associated with a chaperone in the ADP-bound state). With this notation, the rates corresponding to every reaction are easily determined. Examples of the reactions that need to be considered are

- binding/unbinding

$$(0,T,0,0,0,0) \underset{k_{\mathrm{off}}^{\mathrm{adp}}}{\overset{[Hsp70 \cdot ADP]\, k_{\mathrm{on}}^{\mathrm{adp}}\, e^{-\beta \Delta\Delta G}}{\rightleftarrows}} (0,T,0,D,0,0)$$

$$(0,T,0,0,0,0) \underset{k_{\mathrm{off}}^{\mathrm{atp}}}{\overset{[Hsp70 \cdot ADP]\, k_{\mathrm{on}}^{\mathrm{atp}}\, e^{-\beta \Delta\Delta G}}{\rightleftarrows}} (0,T,0,T,0,0)$$

- hydrolysis/synthesis

$$(0,T,0,T,0,0) \underset{k_s^s}{\overset{k_h^s}{\rightleftarrows}} (0,T,0,D,0,0)$$

- nucleotide exchange

$$(0,T,0,D,0,0) \underset{k_{\mathrm{ex,TD}}^{\mathrm{eff}}}{\overset{k_{\mathrm{ex,DT}}^{\mathrm{eff}}}{\rightleftarrows}} (0,T,0,T,0,0).$$

We further provide, as an example, the equation for a precise configuration, say $(0,T,0,D,0,0)$ (here the label stands for the concentration of the configuration). The two binding sites that are

occupied can undergo chaperone unbinding, ATP hydrolysis/synthesis or nucleotide exchange. The remaining unoccupied binding sites can bind either an ATP- or an ADP-bound chaperone. We thus have

$$
\begin{aligned}
\frac{d}{dt}(0,T,0,D,0,0) = \ & -(0,T,0,D,0,0) * (k_{ex,DT}^{eff} + k_s^s + k_{ex,TD}^{eff} + k_h^s + k_{off}^{atp} + k_{off}^{adp}) + \\
& + (0,0,0,D,0,0)[\mathrm{Hsp70 \cdot ATP}]k_{on}^{atp}e^{-\beta\Delta\Delta G} + \\
& + (0,T,0,0,0,0)[\mathrm{Hsp70 \cdot ADP}]k_{on}^{adp}e^{-\beta\Delta\Delta G} + \\
& + (T,T,0,D,0,0)k_{off}^{atp} + \\
& + (D,T,0,D,0,0)k_{off}^{adp} + \\
& + (0,T,T,D,0,0)k_{off}^{atp} + \\
& + (0,T,D,D,0,0)k_{off}^{adp} + \\
& + (0,T,0,D,T,0)k_{off}^{atp} + \\
& + (0,T,0,D,D,0)k_{off}^{adp} + \\
& + (0,T,0,D,0,T)k_{off}^{atp} + \\
& + (0,T,0,D,0,D)k_{off}^{adp} .
\end{aligned}
\tag{11}
$$

Below we further detail the rates of our model.

It is possible to move from an ATP-state to an ADP-state either via hydrolysis/synthesis or via nucleotide exchange. In the case of exchange, effective constants are used, which take into account the unbinding of one nucleotide species and the binding of the different one. The effective exchange rates are thus a function of the ratio [ATP]/[ADP] (see also *De Los Rios and Barducci, 2014*):

$$
k_{ex,DT}^{eff} = \frac{k_{-D}k_{+T}\frac{[ATP]}{[ADP]}}{k_{+D} + k_{+T}\frac{[ATP]}{[ADP]}}
\tag{12}
$$

$$
k_{ex,TD}^{eff} = \frac{k_{-T}k_{+D}}{k_{+D} + k_{+T}\frac{[ATP]}{[ADP]}} ,
\tag{13}
$$

where $k_{+D}$, $k_{+T}$, $k_{-D}$ and $k_{-T}$ are the binding and unbinding rates for ADP and ATP respectively.

The rates of binding between the chaperone and single peptides have been previously determined experimentally (*Mayer et al., 2000*), and they were corrected in order to take into account the conformational change of the full polypeptide substrate upon binding, as we illustrated in the main text.

Substrate binding enhances the chaperone ATPase activity. Furthermore, the stimulation of ATP hydrolysis always takes place in cooperation with JDP co-chaperones. In our model, we did not consider them explicitly but their contribution was implicitly included through the choice of the rate constants.

In particular, the hydrolysis rate in the absence of the substrate, $k_h$, is much smaller than the same rate in the presence of the substrate, $k_h^s$ ($k_h \ll k_h^s$). We assumed that the ratio between the rate of hydrolysis $k_h$ and the rate of synthesis $k_s$ is not altered by the substrate:

$$
\frac{k_h}{k_s} = \frac{k_h^s}{k_s^s} .
\tag{14}
$$

The substrate binding/unbinding rates, the rates of nucleotide exchange and the hydrolysis and synthesis rates are collectively constrained by thermodynamic relations. Indeed, when the ratio between the concentrations of ATP and ADP is equal its equilibrium value (when the spontaneous hydrolysis and synthesis reactions are at steady state and compensate each other), detailed balance must be satisfied (*Ge et al., 2012*). As a consequence, for every closed cycle in the reaction network the product of the rates in one direction must be equal to the product of the rates in the opposite direction. Therefore, if $k_{on}^{atp}$, $k_{on}^{adp}$, $k_{off}^{atp}$ and $k_{off}^{adp}$ are the rate of substrate binding and unbinding from a chaperone in the ATP and ADP states, we must have

$$\frac{k_{on}^{atp}k_h^s k_{off}^{adp}k_s}{k_{on}^{adp}k_s^s k_{off}^{atp}k_h} = \frac{k_{on}^{atp}k_{off}^{adp}}{k_{on}^{adp}k_{off}^{atp}} = 1. \tag{15}$$

Remarkably, taking the rates as provided in *Mayer et al. (2000)*; *Hu et al. (2006)*; *Kellner et al. (2014)*, this relation is not satisfied, and we had thus to modify them. We thus calculated the product in the formula above and then corrected the rates in the following way:

$$\frac{k_{on}^{atp}k_{off}^{adp}}{k_{on}^{adp}k_{off}^{atp}} = r \tag{16}$$

$$k_{on}^{atp}, k_{off}^{adp} \rightarrow k_{on}^{atp}/r^{1/4}, k_{off}^{adp}/r^{1/4} \tag{17}$$

$$k_{on}^{adp}, k_{off}^{atp} \rightarrow k_{on}^{adp}*r^{1/4}, k_{off}^{atp}*r^{1/4}. \tag{18}$$

The concentration of free chaperones in the ATP and in the ADP states was obtained, at the leading order, by solving a three-state system whose reactions have the form

$$\mathrm{Hsp70 + ADP \rightleftharpoons Hsp70 \cdot ADP \rightleftharpoons Hsp70 \cdot ATP \rightleftharpoons Hsp70 + ATP}. \tag{19}$$

Since we worked in the assumption of excess of chaperones in the system, once these concentrations were obtained, they remained fixed once for all, without being considered as a variable of the biochemical network.

We report in the following table the rates used in the model.

**Parameters of the model (*Mayer et al., 2000; Hu et al., 2006; Kellner et al., 2014*)**

| | | | |
|---|---|---|---|
| $k_{off}^{atp}$ | $2.31\ s^{-1}$ | $k_{off}^{adp}$ | $2*10^{-3}\ s^{-1}$ |
| $k_{-T}$ | $1.33*10^{-4}\ s^{-1}$ | $k_{-D}$ | $0.022\ s^{-1}$ |
| $k_{on}^{atp}$ | $1.28*10^{6}\ M^{-1}s^{-1}$ | $k_{on}^{adp}$ | $10^{3}\ M^{-1}s^{-1}$ |
| $k_{+T}$ | $1.3*10^{5}\ M^{-1}s^{-1}$ | $k_{+D}$ | $2.67*10^{5}\ M^{-1}s^{-1}$ |
| $k_h$ | $6*10^{-4}\ s^{-1}$ | $k_h^s$ | $1.8\ s^{-1}$ |

To test the robustness of the model for the radius of gyration, the average FRET efficiency and the free-energy $\Delta G_{swell}$, 100 realizations were implemented, taking each time the values $\Delta G_i$ from a Gaussian distribution with $\sigma = 0.3$ kcal/mol and mean equal to the value obtained with the MD simulations.

## Molecular graphics

Molecular graphics in *Figures 1* and *6* have been generated with UCSF Chimera, developed by the Resource for Biocomputing, Visualization, and Informatics at the University of California, San Francisco, with support from NIH P41-GM103311 (*Pettersen et al., 2004*).

## Additional information

### Funding

| Funder | Grant reference number | Author |
|---|---|---|
| Agence Nationale de la Recherche | ANR-14-ACHN-0016 | Alessandro Barducci |
| Schweizerischer Nationalfonds zur Förderung der Wissenschaftlichen Forschung | 200020_163042 | Paolo De Los Rios |

The funders had no role in study design, data collection and interpretation, or the decision to submit the work for publication.

### Author contributions
Salvatore Assenza, Alberto Stefano Sassi, Validation, Investigation, Methodology; Ruth Kellner, Validation, Investigation; Benjamin Schuler, Resources, Methodology; Paolo De Los Rios, Resources, Supervision, Methodology; Alessandro Barducci, Conceptualization, Resources, Supervision, Methodology, Project administration

### Author ORCIDs
Salvatore Assenza ◍ https://orcid.org/0000-0001-9983-8927
Alberto Stefano Sassi ◍ https://orcid.org/0000-0002-1269-4746
Benjamin Schuler ◍ http://orcid.org/0000-0002-5970-4251
Paolo De Los Rios ◍ https://orcid.org/0000-0002-5394-5062
Alessandro Barducci ◍ https://orcid.org/0000-0002-1911-8039

### Decision letter and Author response
Decision letter https://doi.org/10.7554/eLife.48491.sa1
Author response https://doi.org/10.7554/eLife.48491.sa2

## Additional files
### Supplementary files
• Source data 1. Auxiliary files for molecular dynamics simulations.

• Transparent reporting form

### Data availability
All the source data used for generating relevant figures (Figures 1, 2, 4, 5, 6, Figure 2—figure supplement 1, Appendix 1—figure 1) have been provided as supporting files. All the information necessary for reproducing the molecular simulations have been deposited in github (https://github.com/saassenza/Hsp70Unfoldase; also provided as Source data 1) and PLUMED NEST (plumID:19.076) repositories.

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

## Appendix 1

### Limited role of electrostatic interactions in chaperone/ substrate complexes

The choice of neglecting electrostatic interactions in the CG simulation of DnaK/rhodanese complexes was motivated by the results of two control experiments, which were performed exactly at the same conditions except for the salt concentration, namely 5 mM and 100 mM KCl. A double-cysteine variant of rhodanese (K135C/K174C), which was produced by site-directed mutagenesis and prepared as described before for wild-type rhodanese (*Miller et al., 1992*), was labeled with Alexa Fluor 488 C5 maleimide and Alexa Fluor 594 C5 maleimide (Invitrogen, Molecular Probes) (*Hillger et al., 2007*). A twofold molar excess of the dyes was added to the protein and incubated for 1 hr at room temperature. Unreacted dye was removed by gel filtration followed by anion exchange chromatography to reduce the amount of incorrectly labelled protein using a MonoQ 5/50 GL column (GE Healthcare) installed on an ÄKTA purifier FPLC system equilibrated in 50 mM Tris·HCl, pH 7.0, and eluted with a gradient from 0 to 500 mM sodium chloride over 60 mL (12 column volumes). The chaperone proteins DnaK and DnaJ (stock solution concentration 100 µM in 50 mM Tris HCl, pH 7.7, 100 mM NaCl) were gifts from H.-J. Schönfeld (Hoffmann-La Roche Ltd., Basel). Labelled rhodanese was denatured in 4 M guanidinium chloride in buffer (50 mM Tris HCl, 10 mM MgCl$_2$, 200 mM β-mercaptoethanol, and 0.001% Tween 20) with either 5 or 100 mM KCl added. The denatured rhodanese was diluted 100x into buffer containing 10 µM DnaK, 0.5 µM DnaJ, 1 mM ATP and 5 or 100 mM KCl to form chaperone rhodanese complexes at a final concentration of 50 pM rhodanese. Single-molecule Förster Resonance Energy Transfer (FRET) measurements were started immediately after dilution and data recorded at 22°C for 30 min to construct the FRET efficiency histograms. Data were recorded with a MicroTime 200 confocal microscope (PicoQuant) and on a custom-built confocal microscope. All measurements were obtained with pulsed interleaved excitation (*Müller et al., 2005*). The instrument set up and data reduction were the same as described before (*Kellner et al., 2014*). The collected FRET histograms are plotted in *Appendix 1—figure 1*. The average FRET efficiencies were 0.38 ± 0.01 with 5 mM KCl and 0.40 ± 0.01 with 100 mM KCl (errors indicate uncertainties estimated from the standard deviation of the larger data set collected at 5 mM salt). The very similar values obtained at different salt concentrations suggest that electrostatic interactions do not play a significant role in determining the conformational properties of rhodanese/chaperone complexes.

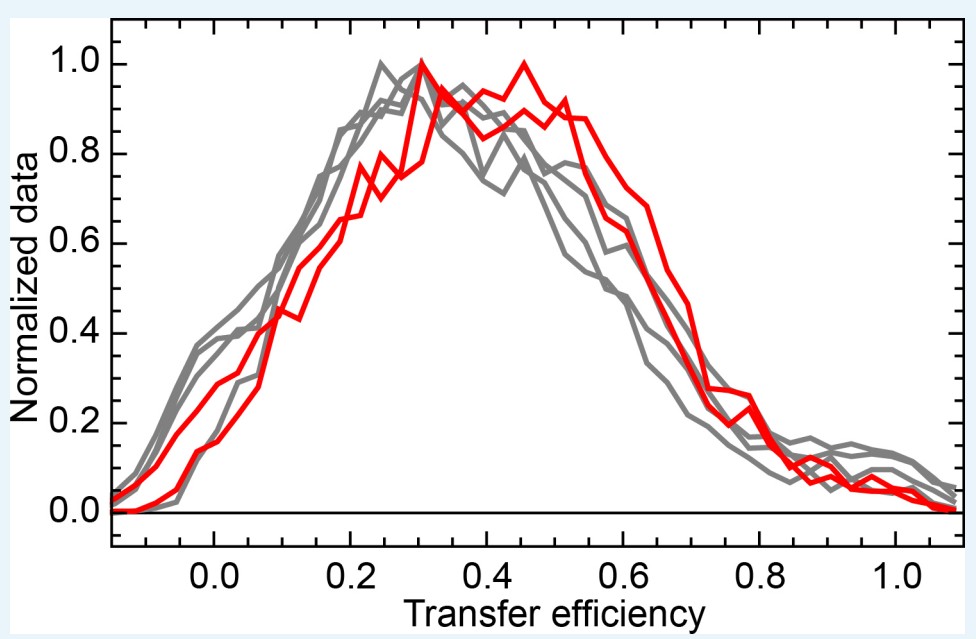

**Appendix 1—figure 1.** Normalized FRET efficiency histograms of the variant Δ39 (K135C/K174C). Experiments were performed in the presence of 10 μM DnaK, 500 nM DnaJ, 1 mM ATP and either 5 mM (grey curves) or 100 mM (red curves) of KCl.

The online version of this article includes the following source data is available for figure 1:

**Appendix 1—figure 1—source data 1.** Experimental data employed to generate the FRET histograms reported in the *Appendix 1—figure 1*.

## Appendix 2

### Sanchez theory for coil-to-globule transition

In his theory on polymer coil-to-globule transition **Sanchez (1979)** considers a Freely-Jointed Chain (FJC) made of $n$ monomers characterized by attractive interactions of average magnitude $\epsilon$. Let $\alpha \equiv R_g/R_{g,0}$ be the expansion parameter, where $R_g$ is the radius of gyration of the polymer, while $R_{g,0}$ is the value of $R_g$ in the unperturbed case ($\epsilon = 0$ and no excluded volume present). Assuming a Flory-Fisk distribution for the unperturbed case, the probability distribution of the expansion parameter $P(\alpha)$ was shown to be (**Sanchez, 1979**)

$$P(\alpha) = \frac{1}{Z}\alpha^6 e^{-\frac{7}{2}\alpha^2 + nq(\epsilon,\alpha)}\,, \tag{20}$$

where $Z$ is the partition function and

$$q(\epsilon,\alpha) = \frac{1}{2}\epsilon\frac{\phi_0}{\alpha^3} - \left(\frac{\alpha^3}{\phi_0} - 1\right)\ln\left(1 - \frac{\phi_0}{\alpha^3}\right)\,,$$

with $\phi_0 = \sqrt{19/(27n)}$. The free energy can then be straightforwardly computed as $\Delta G(\alpha) = -k_B T \ln P(\alpha)$, that is making the dependence on $R_g$ explicit,

$$\Delta G(R_g) = -k_B T \ln P(R_g/R_{g,0})\,. \tag{21}$$

In the present case, rhodanese is constituted by 293 amino acids, that is there are $N_b = 292$ bonds. The bond length is $b_l = 3.8$ Å, estimated as the typical distance between $C_\alpha$ atoms belonging to consecutive residues (**Hofmann et al., 2012**). In order to apply the Sanchez theory, one needs to consider the FJC equivalent to rhodanese. To this aim, the Kuhn length $b_K$ can be estimated as $b_K = 2l_p = 8$ Å, where $l_p = 4$ Å is the persistence length of an unfolded protein under native conditions (**Hofmann et al., 2012**). The number of monomers $n$ in the equivalent FJC can then be computed by imposing the total contour length of rhodanese: $n = N_b b_l/b_K$. From there, both $\phi_0$ and the unperturbed radius of gyration $R_{g,0} = \sqrt{nb_K^2/6}$ can be calculated, leaving $\epsilon$ as the only unknown quantity in **Equation (20)**. The latter can be fixed by imposing that $\Delta G$ has a minimum at the experimental value $R_g^* = 20.1$ Å obtained in the absence of chaperones (**Kellner et al., 2014**; **Hofmann et al., 2014**), yielding $\epsilon \simeq 1.07$ kcal/mol. Plugging the values of the parameters derived above into **Equation (21)** gives the black line reported in **Figure 2** in the main text, in excellent agreement with the simulation results. We stress that neither the force field nor the MD results were employed in the derivation of the parameters used **Equation (20)**.

