## [Decision Letter]

**Acceptance summary:**

In this paper, the authors provide a very nice description of how non-equilibrium effects can drive a particular biological phenomenon.

**Decision letter after peer review:**

Thank you for submitting your article "Efficient conversion of chemical energy into mechanical work by Hsp70 chaperones" for consideration by *eLife*. Your article has been reviewed by two peer reviewers, and the evaluation has been overseen by Arup Chakraborty as the Senior and Reviewing Editor. The following individual involved in review of your submission has agreed to reveal their identity: Elena Papaleo (Reviewer #2).

The reviewers have discussed the reviews with one another and the Reviewing Editor has drafted this decision to help you prepare a revised submission.

Summary:

Your paper, which builds on a previous publication, provides new insights into a non-equilibrium biological problem, and also represents a nice use of MD calculations coupled to Jarzinsky's relation. There are two major comments that need to be addressed.

Essential revisions:

1) The kinetic rate model is very simple, however with many parameters. The innovation seems to be that you incorporate the free energies ∆∆G (calculated from MD) into the kinetic rate model, and obtain the probabilities of n. If we understood correctly, the end result that higher the concentration of ATP, higher the occupation number *n* seems simple enough, because you choose konATP≫konADP in the table of parameters, and therefore one should naturally expect it. So, it is unclear what is non-trivial in this interpretation given the choice konATP≫konADP.

2) The paper is not written very clearly, especially for a general biological audience, which constitutes the readership of *eLife*.

We have appended the two reviews below, so you can directly address other points that need to be addressed.

*Reviewer #1:*

The authors study the probability of binding of chaperones to protein substrates. The article utilizes (i) MD simulations towards calculating free energy differences (∆∆*G*) of incremental binding of chaperon complexes onto the substrate, (ii) uses a simple polymer model to predict the "free energy differences" (∆*G*), and (iii) implement a multi-parameter kinetic rate model to predict the probability of number of occupied chaperones as a function of ratio of ATP and ADP concentrations. These studies seem to be extensions of some of the authors' previous studies on chaperones, as they utilize some parameters from their earlier studies.

1) The article utilizes Jarzynski's equality for calculating the free energy differences. The free energy calculations are very expensive in nature due to the possibility of many conformations and number of states. The calculations appear fine. The agreement between polymer theory and the MD calculations is nice. The calculations by taking the stretched end state as a reference is clever. This work provides another example for the utility of non-equilibrium work calculations in obtaining the free energies for biological systems. I liked this aspect of the work.

2) Here is my main comment: The kinetic rate model is very simple, however with many parameters. The innovation seems to be that they incorporate the free energies ∆∆*G* (calculated from MD) into the kinetic rate model, and obtain the probabilities of n. If I did not misunderstand, the end result that higher the concentration of ATP, higher the occupation number *n* seems simple enough, because they choose konATP≫konADP in the table of parameters, and therefore one should naturally expect it. So, I would like to know what is non-trivial in this interpretation given the choice konATP≫konADP.

A far more interesting study would have been to understand the mechanisms that help us understand konATP≫konADP depending on the protein and chaperon configurations and allostery, but this is beyond the scope of the current article. So, the authors may take this as a suggestion for future works.

3) Also, the authors arguments on energy efficiencies in converting the chemical energy to mechanical energy are not motivated properly, as the choice for the definition of efficiency is not clear. Why is Equation 2 a good definition for efficiency? It’s just that the system is not in thermal equilibrium and because of the rates chosen, one will expect to see increased probabilities of occupation of chaperones with increased concentration ratio of ATP to ADP?

Decision: I am inclined to consider the article for publication in *eLife* when I see the response for point 2.

*Reviewer #2:*

The work by Barducci's group and collaborators illustrates an elegant computational study, validated against experimental data, to unveil thermodynamics and kinetic details of the mechanism of substrate expansion induced by the Hsp70 chaperone.

The work is technically sound and well designed, the agreement with the FRET data are encouraging in the direction that this kind of models can provide important insight into the mechanistic aspects of the process and the critical role played by ATP.

Despite the impressive work, the presentation of the data in the manuscript can be improved and guides the reader better in conveying some of the main messages. I thus encourage the authors to revise the writing and data presentation in figures and supplementary information carefully. Sometimes the writing sounds rather technical. A revision could be done to make it more accessible to a broader readership, especially more biology-oriented so that the outcome of this nice work can be better appreciated not only by specialists and does not risk to get lost.

Subsection “Structural and thermodynamic characterisation of chaperone-substrate complexes” – the authors say that they used a coarse-grained force field tailored to disordered proteins and it works appropriately for the unfolded state of the substrate of interest, but are they using the same model also for the chaperone protein? How did they ensure that this model is a good compromise to simulate at the same time the unfolded substrate and the folded chaperone? Could the authors discuss more in details about this?

Discussion – how much it is expected to be a specific mechanism for Hsp70 or could be common to other chaperones? Perhaps the authors could consider having a more general discussion and future perspective.

Another concern that I have is about data availability to reproduce the calculations/simulations and comparisons with experimental data. There are different options that the authors can consider and widely used by the molecular modeling and computational community. For example, they could use a GitHub repository to release the input files for the simulations, scripts to reproduce them, along with scripts to reproduce the analyses. GitHub, of course, has limitation to raw data if the authors want to make accessible the trajectories, they should probably look for repositories that provide more space than GitHub. Otherwise, they could add a statement in the manuscript that the trajectories will be available upon request, as it has been done in other similar publications. I believe that the input files and scripts would already be a significant advantage in the direction of accessibility and reproducibility.

The corresponding author is one of the members of the recently established PLUMED consortium, and I would encourage to deposit the input files also in the PLUMED-NEST repository.

---

## [Author Response]

Essential revisions:1) The kinetic rate model is very simple, however with many parameters. The innovation seems to be that you incorporate the free energies ∆∆G (calculated from MD) into the kinetic rate model, and obtain the probabilities of n. If we understood correctly, the end result that higher the concentration of ATP, higher the occupation number n seems simple enough, because you choose konATP≫konADP in the table of parameters, and therefore one should naturally expect it. So, it is unclear what is non-trivial in this interpretation given the choice konATP≫konADP.

We acknowledge the reviewer for this comment that provides us the opportunity to clarify a key point of our study. Actually, an excess of ATP over ADP in absence of any Hsp70-induced ATPase activity would trivially correspond to an excess of ATP-bound Hsp70s over ADP-bound chaperone. In this scenario, the effective affinity of the chaperones would actually decrease, since Hsp70 is actually more affine to the substrate in ADP- than in ATP-bound state, as suggested by the dissociation constants (*Kd*) of the corresponding chaperone/substrate complexes (koffATP/konATP>koffADP/konADP), regardless of the kinetic binding constants (konATP≫konADP). The “ultra-affine” binding of Hsp70 to substrate in excess of ATP over ADP is indeed a non-equilibrium effect that strictly depends on the hydrolysis of ATP molecules in the Hsp70-substrate complex and the consequent conformational change of the chaperone. To illustrate this key feature, in the revised manuscript we modified Figure 4 to show the dependence of the average occupation number <*n*> on ATP/ADP ratio both in equilibrium and non-equilibrium scenarios and we added a short discussion: “It is important here to underscore again that the binding of the chaperones in these conditions is a non-equilibrium effect, driven by the Hsp70-induced hydrolysis of ATP, and it is not a mere consequence of the excess of ATP over ADP or of the large difference between the substrate association rates to the ATP- and ADP-bound states. […] As a matter of fact, in such equilibrium scenario an excess of ATP over ADP actually slightly disfavors chaperone binding, because the Hsp70 affinity for the substrate is slightly lower in the ATP-bound state than in the ADP-bound state (koffATP/konATP>koffADP/konADP), (see De Los Rios and Barducci, 2014 for further discussion.).”

2) The paper is not written very clearly, especially for a general biological audience, which constitutes the readership of eLife.

We extensively revised the manuscript in order to increase its clarity and readability. Particularly, we followed the suggestions of reviewer 2 and we substantially enriched the Introduction and the Discussion sections to provide a more general background and to illustrate the implications of our results in a broader context. Furthermore, we revised a few key points in the Results section to improve readability and we completely reshaped the Supplementary Information into an extended Materials and methods section and two appendices. We are confident that our work is now accessible to the large community of researchers interested in the physics of living system or quantitative biology.

We have appended the two reviews below, so you can directly address other points that need to be addressed.Reviewer #1:The authors study the probability of binding of chaperones to protein substrates. The article utilizes (i) MD simulations towards calculating free energy differences (∆∆G) of incremental binding of chaperon complexes onto the substrate, (ii) uses a simple polymer model to predict the "free energy differences" (∆G), and (iii) implement a multi-parameter kinetic rate model to predict the probability of number of occupied chaperones as a function of ratio of ATP and ADP concentrations. These studies seem to be extensions of some of the authors' previous studies on chaperones, as they utilize some parameters from their earlier studies.1) The article utilizes Jarzynski's equality for calculating the free energy differences. The free energy calculations are very expensive in nature due to the possibility of many conformations and number of states. The calculations appear fine. The agreement between polymer theory and the MD calculations is nice. The calculations by taking the stretched end state as a reference is clever. This work provides another example for the utility of non-equilibrium work calculations in obtaining the free energies for biological systems. I liked this aspect of the work.2) Here is my main comment: The kinetic rate model is very simple, however with many parameters. The innovation seems to be that they incorporate the free energies ∆∆G (calculated from MD) into the kinetic rate model, and obtain the probabilities of n. If I did not misunderstand, the end result that higher the concentration of ATP, higher the occupation number n seems simple enough, because they choose konATP≫konADP in the table of parameters, and therefore one should naturally expect it. So, I would like to know what is non-trivial in this interpretation given the choice konATP≫konADP.

We addressed this comment above in our response to the “essential revision”.

A far more interesting study would have been to understand the mechanisms that help us understand konATP≫konADP depending on the protein and chaperon configurations and allostery, but this is beyond the scope of the current article. So, the authors may take this as a suggestion for future works.

We thank the reviewer for her/his suggestion. Actually, the available structures of ATP- and ADP-bound Hsp70 already provide crucial molecular details that greatly help in rationalizing the kinetic properties of chaperone-peptide complexes. We therefore revised the Introduction section to better clarify this point by adding the following paragraph “More precisely, when the chaperone is in the ATP-bound state, the SBD is open and easily accessible to the substrate, whereas the SBD is closed when ADP is bound. These structural differences result in substrate binding and unbinding rates when ATP is bound that are orders of magnitude faster than when ADP is bound”

3) Also, the authors arguments on energy efficiencies in converting the chemical energy to mechanical energy are not motivated properly, as the choice for the definition of efficiency is not clear. Why is Equation 2 a good definition for efficiency? It’s just that the system is not in thermal equilibrium and because of the rates chosen, one will expect to see increased probabilities of occupation of chaperones with increased concentration ratio of ATP to ADP?

As we stressed above, the efficient Hsp70 binding, and the consequent mechanical work, observed in excess of ATP strictly depends on the ATP hydrolysis catalysed by the chaperones and it cannot be trivially explained by the concentrations of ATP- and ADP-bound Hsp70. In this scenario, we measured the effectiveness of the overall process by considering the ratio between the conformational free-energy difference due to Hsp70 binding (Δ*G_Swell_*, defined Equation 2) and the free energy associated to ATP hydrolysis (Δ*G_h_*, now explicitly defined in Equation 3). We believe that this is a natural choice to evaluate how effectively Hsp70 can convert the chemical energy of ATP (Δ*G_h_*) into mechanical work (Δ*G_Swell_*).

Decision: I am inclined to consider the article for publication in eLife when I see the response for point 2.Reviewer #2:The work by Barducci's group and collaborators illustrates an elegant computational study, validated against experimental data, to unveil thermodynamics and kinetic details of the mechanism of substrate expansion induced by the Hsp70 chaperone.The work is technically sound and well designed, the agreement with the FRET data are encouraging in the direction that this kind of models can provide important insight into the mechanistic aspects of the process and the critical role played by ATP.Despite the impressive work, the presentation of the data in the manuscript can be improved and guides the reader better in conveying some of the main messages. I thus encourage the authors to revise the writing and data presentation in figures and supplementary information carefully. Sometimes the writing sounds rather technical. A revision could be done to make it more accessible to a broader readership, especially more biology-oriented so that the outcome of this nice work can be better appreciated not only by specialists and does not risk to get lost.Subsection “Structural and thermodynamic characterisation of chaperone-substrate complexes” – the authors say that they used a coarse-grained force field tailored to disordered proteins and it works appropriately for the unfolded state of the substrate of interest, but are they using the same model also for the chaperone protein? How did they ensure that this model is a good compromise to simulate at the same time the unfolded substrate and the folded chaperone? Could the authors discuss more in details about this?

We thank the reviewer for her comment that gives us the opportunity to improve the clarity of the manuscript. We actually used a CG force-field devised for disordered protein to simulate the unfolded substrate whereas for Hsp70 we relied on a simple structure-based CG model that was meant to account only for chaperone-chaperone and chaperone-substrate excluded-volume interaction. In the revised manuscript, we explained this key point both in the Results and in the Materials and methods section.

Discussion – how much it is expected to be a specific mechanism for Hsp70 or could be common to other chaperones? Perhaps the authors could consider having a more general discussion and future perspective.

We thank the reviewer for this precious suggestion. We now included in the discussions some considerations about the significance of our findings trying to make contact with recent results in the general context of cellular proteostasis. “From a broader perspective, the ATP-driven action of Hsp70s induces a *non-equilibrium* redistribution of their protein-substrates over their structural ensemble. In particular, thanks to the fine-tuning of the process by co-chaperones (J-domain proteins and Nucleotide Exchange Factors), the expansion process highlighted here, followed by substrate release, may result in the enhancement of the native state population beyond the predictions of thermodynamic equilibrium, as recently observed even under otherwise denaturing conditions (Goloubinoff et al., 2018). […] Likewise, they raise fundamental questions about the evolution of protein sequences: indeed, since chaperones are ubiquitous and very much conserved across the different kingdoms of life, their ability to favor native states might have partially relieved the selection pressure for strong *equilibrium* thermodynamic stability, thus allowing evolution to proceed faster and to be more tolerant for slightly destabilizing mutations, as suggested in Rutherford and Lindquist, 1998, Tokuriki and Tawfik, 2009”.

Another concern that I have is about data availability to reproduce the calculations/simulations and comparisons with experimental data. There are different options that the authors can consider and widely used by the molecular modeling and computational community. For example, they could use a GitHub repository to release the input files for the simulations, scripts to reproduce them, along with scripts to reproduce the analyses. GitHub, of course, has limitation to raw data if the authors want to make accessible the trajectories, they should probably look for repositories that provide more space than GitHub. Otherwise, they could add a statement in the manuscript that the trajectories will be available upon request, as it has been done in other similar publications. I believe that the input files and scripts would already be a significant advantage in the direction of accessibility and reproducibility.The corresponding author is one of the members of the recently established PLUMED consortium, and I would encourage to deposit the input files also in the PLUMED-NEST repository.

All the relevant information for reproducing the simulation results are now deposited on GitHub (https://github.com/saassenza/Hsp70Unfoldase) and in the PLUMED NEST (plumID:19.076) repositories.